# Lnc*PLAAT3*-AS Regulates *PLAAT3*-Mediated Adipocyte Differentiation and Lipogenesis in Pigs through miR-503-5p

**DOI:** 10.3390/genes14010161

**Published:** 2023-01-06

**Authors:** Zhiming Wang, Jin Chai, Yuhao Wang, Yiren Gu, Keren Long, Mingzhou Li, Long Jin

**Affiliations:** 1Key Laboratory of Livestock and Poultry Multiomics, Ministry of Agriculture and Rural Affairs, College of Animal Science and Technology, Sichuan Agricultural University, Chengdu 611130, China; 2Sichuan Key Laboratory of Animal Breeding and Genetics, Sichuan Institute of Animal Science, Chengdu 610066, China; 3Sichuan Provincial Key Laboratory of Animal Breeding and Genetics, Institute of Animal Genetics and Breeding, Sichuan Agricultural University, Chengdu 611130, China

**Keywords:** primary adipocyte cells, adipose tissue, lncRNA-AS, RNA-seq

## Abstract

Animal fat deposition has a significant impact on meat flavor and texture. However, the molecular mechanisms of fat deposition are not well understood. Lnc*PLAAT3*-AS is a naturally occurring transcript that is abundant in porcine adipose tissue. Here, we focus on the regulatory role of lnc*PLAAT3*-AS in promoting preadipocyte proliferation and adipocyte differentiation. By overexpressing or repressing lnc*PLAAT3* expression, we found that lnc*PLAAT3*-AS promoted the transcription of its host gene *PLAAT3*, a regulator of adipocyte differentiation. In addition, we predicted the region of lnc*PLAAT3*-AS that binds to miR-503-5p and showed by dual luciferase assay that lnc*PLAAT3*-AS acts as a sponge to absorb miR-503-5p. Interestingly, miR-503-5p also targets and represses *PLAAT3* expression and helps regulate porcine preadipocyte proliferation and differentiation. Taken together, these results show that lnc*PLAAT3*-AS upregulates *PLAAT3* expression by absorbing miR-503-5p, suggesting a potential regulatory mechanism based on competing endogenous RNAs. Finally, we explored lnc*PLAAT3*-AS and *PLAAT3* expression in adipose tissue and found that both molecules were expressed at significantly higher levels in fatty pig breeds compared to lean pig breeds. In summary, we identified the mechanism by which lnc*PLAAT3*-AS regulates porcine preadipocyte proliferation and differentiation, contributing to our understanding of the molecular mechanisms of lipid deposition in pigs.

## 1. Introduction

In recent years, genome annotation has identified many lncRNAs, although the functions of most of these lncRNAs are still unknown [1]. Some antisense lncRNAs have been demonstrated to have functions. For example, the *ZFPM2* antisense RNA 1 (*ZFPM2*-AS1) lncRNA has been reported to regulate the migration and invasion of hepatocellular carcinoma cells by mediating the miR-139/GDF10 axis [2]. *GAS6* antisense RNA 1 (*GAS6*-AS1) regulates cell proliferation and invasion in clear cell renal cell carcinoma (*ccRCC*) by mediating the AMPK/mTOR signaling pathway, suggesting that *GAS6-*AS1 may be a potential therapeutic target in *ccRCC* [3]. *Nqo1*-AS1 upregulates Nqo1 expression by binding to the *Nqo1* 3′UTR and increasing *Nqo1* mRNA stability, thereby attenuating cigarette smoke-induced oxidative stress [4]. Although many studies have been conducted on antisense lncRNAs [5], the role of lncRNAs in porcine adipogenesis remains largely unknown.

Pigs are an essential animal in the agricultural economy and an important source of meat worldwide [6]. In addition, domestic pigs are essential model animals that are frequently used in medical research due to having a similar genome size, gene structure and function, anatomical structure of the digestive organs, metabolic patterns, visceral organ metabolism, and dietary habits (omnivorous diet) to humans [7,8]. In nutritional, metabolic, and cardiovascular studies, as well as in several other areas of biomedical research, pigs have proven to be valuable animal models [9].

It has been previously reported that the knockdown of *PLAAT3* significantly inhibits lipid deposition in mice [10]. Previous studies have shown that lnc*PLAAT3*-AS enhances *PLAAT3* mRNA stability by forming an RNA–RNA dimer with the *PLAAT3* transcript [11]. However, there is no direct evidence of a specific regulatory role for lnc*PLAAT3*-AS in adipogenesis or for the associated molecular mechanisms. To analyze the role of lnc*PLAAT3*-AS in lipogenic differentiation, we overexpressed or knocked down lnc*PLAAT3*-AS in porcine primary preadipocytes. The results from these experiments showed that lnc*PLAAT3*-AS regulates the expression of cell cycle-related genes and promotes the differentiation of preadipocytes. lnc*PLAAT3*-AS promoted adipocyte differentiation by acting as a sponge for miR-503-5p, thereby repressing the activity of this miRNA and promoting the increased expression of *PLAAT3*. In conclusion, our findings demonstrate the molecular mechanism by which a lncRNA regulates adipogenesis and provide a potential molecular approach to improving lean muscle mass in livestock production by inhibiting fat deposition.

## 2. Materials and Methods

### 2.1. Ethics Statement

All procedures involving animals in this study were managed and operated in strict accordance with the Regulations for the Management of Laboratory Animal Affairs (Ministry of Science and Technology, Beijing, China, June 2004) and were also approved by the Animal Care and Use Group Institution, College of Animal Science and Technology, Sichuan Agricultural University, China, under license number NO.20210162.

### 2.2. Cell Culture and Differentiation

The piglet was humanely killed and the fat tissue from the backs of the female Rongchang piglets was collected to obtain preadipocytes. After removing the soft tissue and small tubes, the remaining fat tissue was washed three times with phosphate-buffered salt water (PBS, Gibco, Carlsbad, CA, USA). Next, about 60 g of the fat tissue was quickly minced and digested at 37 °C for 1 h with 0.25% type I collagenase (Invitrogen, Carlsbad, CA, USA). The digested tissue solution was then centrifuged at 1000 rpm for 8 min. The cells were passed through 100 um strainers, and the pellet (containing cleaned preadipocytes) was resuspended in Dulbecco’s changed Eagle medium (Gibco, Carlsbad, CA, USA) containing 20% fetal calf serum (Gibco, Carlsbad, CA, USA). The preadipocytes were cultured containing 5% CO^2^ and maintained at 37 °C. When preadipocytes reached 80% confluence, they were digested with trypsin and passaged in new neo-culture plates. After the preadipocytes had grown and spread to an appropriate density in the cell culture plates, they were treated with insulin (Sigma, Saint Louis, MI, USA), 3-isobutyl 1-methylxanthine (Sigma, Saint Louis, MI, USA), and dexamethasone (Sigma, Saint Louis, MI, USA) to induce differentiation of the preadipocytes into mature adipocytes [12]. All cell experiments were repeated three times independently, and the cells used on all three occasions were from the same pig.

### 2.3. Cell Transfection

Cells were transfected using lipofectamine 3000 (Lipo3000, Invitrogen) according to the manufacturer’s instructions, with opti-MEM (Gibco, Carlsbad, CA, USA) as an auxiliary transfection reagent. Taking 6-well plates containing 2 mL of medium per well as an example, 100 µL of opti-MEM was first incubated with 5 µL of Lipo3000, separately, for 5 min. Subsequently, the two solutions were combined and incubated for 10 min, after which the combined solution was added to the culture medium to transfect the cells.

### 2.4. Western Blot Assay

Cells were collected from six-well plates using a cell spatula. The extracted protein was isolated using a radio immunoprecipitation assay (RIPA) kit (Bio-Rad, Hercules, CA, USA). After the proteins were extracted, the BCA kit was used to determine their concentration. Absin-prepared SDS-PAGE gels (Absin Bioscience Inc., Shanghai, China) were used to separate the proteins. Using a BIO-RAD protein transfer machine, the products were then transferred onto polyvinylidene difluoride membranes and incubated with primary antibodies. Protein bands were detected using ECL (Service) and visualized using the ECL Western Blot Detection Kit (Thermo Scientific, Rockford, IL, USA) and the Image Quant LAS 4000 kit (GE HealthCare, Chicago, IL, USA). All western blotting experiments were repeated at least three times.

### 2.5. Luciferase Reporter Assay

Wild type (WT) and MUT lncPLAAT3-AS plasmids were constructed using the pcDNA3.1 vector. They were transfected with miR-503-5p mimics, miR-503-5p inhibitor, miR-503-5p mimics negative control (mimics NC), and miR-503-5p inhibitor negative control (inhibitor NC) using the lipofectamine 3000 reagents into PK15 cells as described in Section 2.4. After 48 h, the samples were collected, and the fluorescence strength was detected using the Dual-Glo Luciferase Assay System (Promega, Madison, WI, USA) following the manufacturer’s instructions.

### 2.6. Assessment of Cell Proliferation via CCK-8 and EdU Assays

The growth and spread of cells were detected using the Cell Counting Kit 8 (CCK-8, Biosharp, Hefei, China) and 5-ethynyl-20-deoxyuridine (EdU, Ribobio, Guangzhou, China). After transfection, 10 μL of CCK-8 reagent was added to each well of the cell culture plate, and the absorbance at 450 nm was measured after 24 h. Preadipocytes were seeded in 12-well plates for the EdU assay. Following transfection, 100 mL of 50 mM EdU reagent was added to each well and grown for at least 24 h before images were taken with a Nikon TE2000 microscope (Nikon, Tokyo, Japan).

### 2.7. RNA Isolation and Reverse Transcription

Trizol (Invitrogen) was used to extract the total RNA from cells. The total RNA was reverse transcribed to cDNA using the HiScript III RT Super Mix (Vazyme, Nanjing, China), oligo-dTs according to the manufacturer’s instructions. Denaturing gel electrophoresis and spectrophotometry (Thermo, Waltham, MA, USA) was used to measure RNA mass and concentration, and the final product was diluted to the appropriate volume with water.

### 2.8. Real-Time Quantitative PCR

Taq Pro Universal SYBR qPCR Master Mix reagents (Vazyme, Nanjing, China) were used for real-time quantitative PCR (RT-qPCR). Primer5 software was used to design RT-qPCR primers, and their sequences are shown in Table 1. RT-qPCR was performed on a Bio-Rad CFX96 Real-Time PCR detection system. Each sample was analyzed in triplicate. The relative expression levels of each gene tested within the samples were calculated using the 2^−ΔΔCT^ method.

### 2.9. Oil Red O Staining

After induction of adipogenesis, cells were washed two to three times with PBS (Gibco, Carlsbad, CA, USA) and then fixed for 30 min in 4% paraformaldehyde. The samples were rinsed twice with 60% isopropanol and dried for 30 min before being treated with 1 mL of the oil red O dye working solution. A microscope was used to observe the oil red O staining after adding 1 mL PBS (Gibco, Carlsbad, CA, USA) to the culture plate.

### 2.10. RNA-Seq and Collection of Sequencing Data

Ampure XP beads (Beckman Coulter, Brea, CA, USA) were used to purify the PCR-amplified cDNA fragments with adapters. The switching mechanism at the 5′ end of the RNA transcript primer with oligo(dG) at the 3′ end was added in advance to the cDNA synthesis reaction. The switching mechanism at the 5′ end of the RNA transcript primer oligo(dG) pairs with the protruding C’s at the end of the synthetic cDNA to form an extension template for the cDNA, and the reverse transcriptase automatically switches the template to use the SMART primer as an extension template to continue extending the cDNA strand to the end of the primer. All resulting cDNA strands have an oligo(dT)-containing primer sequence at one end and a known SMART primer sequence at the other end, which can be amplified using universal primers after the second strand has been synthesized. A Bioanalyzer Agilent Technologies 2100 (Agilent Technologies, Santa Clara, CA, USA) was used to validate the cDNA library. Following heat-denatured PCR products, splint oligos were used to circularize them. The final library was considered to be the single-stranded circular DNA. After the final library was amplified with phi29 (Thermo Fisher Scientific, Waltham, MA, USA), over 300 copies of each molecule were produced as DNA nanoballs (DNBs) [13,14]. A Bioanalyzer Agilent Technologies 2100 was used to validate the cDNA library. Following heat-denatured PCR products, splint oligos were used to circularize them. The final library was considered to be the single-stranded circular DNA. After the final library was amplified with phi29 (Thermo Fisher Scientific, Waltham, MA, USA), over 300 copies of each molecule were produced as DNBs. The DNBs loaded into the patterned nanoarray were read using the BGISEQ500 platform (BGI, Shenzhen, China).

FAStQC software (Version 0.11.9) was used to quality control (QC) the raw data obtained from sequencing, and the Q30 values and gas chromatograph (GC) content of the sequencing data were calculated. Hisat software (Version 2 2.2.1) was used to perform genome alignment of the clean reads, and 90.55–91.20% could be aligned to the reference genome, indicating valid reads and good sequencing results. The differentially expressed genes were characterized, and Gene Ontology (GO) and Kyoto Encyclopedia of Genes and Genomes (KEGG) functional enrichment analyses were performed

A total of seven Chinese pig breeds were downloaded, including Chenghua, Neijiang, Tibetan, Qingyu, Wujin, Yacha, and Yanan, and one Western breed, Yorkshire. RNA-seq data were downloaded from the National Library of Medicine database (https://www.ncbi.nlm.nih.gov/geo/, accessed on 20 July 2022, accession numbers SRP090525) [15,16].

### 2.11. Data Analysis and Statistics

At least three independent replicates were performed for each experiment. MRNA and miRNA expression levels were calculated using the 2^−ΔΔCt^ method, and data are presented as the mean ± SEM of each group. ANOVA and Student’s *t*-test were applied to assess the differences in expression levels between groups using GraphPad Prism 8.0 (GraphPad Software), with *p* < 0.05 considered significant and *p* < 0.01 highly significant. A * indicates significant differences, and ** indicates highly significant differences.

## 3. Results

### 3.1. LncPLAAT3-AS Regulates Porcine Primary Preadipocyte Proliferation

To assess the role of lnc*PLAAT3*-AS in porcine primary preadipocyte proliferation, we transfected porcine primary preadipocytes with a lnc*PLAAT3*-AS overexpression plasmid or siRNA-lnc*PLAAT3*-AS. As shown in Figure 1A, lnc*PLAAT3*-AS expression was significantly decreased after transfection with siRNA-lnc*PLAAT3*-AS and significantly increased after transfection with the lnc*PLAAT3*-AS overexpression plasmid compared with the negative control (*p* < 0.01). Next, we performed an EdU and proliferation assay and found that the percentage of EdU-positive cells increased after overexpression of lnc*PLAAT3*-AS, while transfection with siRNA-lnc*PLAAT3*-AS siRNA had the opposite effect. The results from the CCK-8 assay results were similar: the cell fluorescence value at 450 nm increased by 50% after overexpression of lnc*PLAAT3*-AS compared with the negative control (*p* < 0.01) (Figure 1B,D). We then examined the expression of genes involved in cell proliferation and apoptosis by RT-qPCR results and found that *CCND1*, *CCNE1*, and *CDK4* were significantly upregulated after transfection with the lnc*PLAAT3*-AS overexpression plasmid compared with the negative control, while transfection with siRNA-lnc*PLAAT3*-AS had the opposite effect (*p* < 0.01). In addition, lnc*PLAAT3*-AS overexpression resulted in a significant decrease in the expression of cell cycle protein-dependent kinase inhibitor (*p53*), a marker of inhibited cell proliferation (*p* < 0.01) (Figure 1C). Taken together, these results suggest that lnc*PLAAT3*-AS plays a role in promoting the proliferation of porcine primary preadipocytes.

### 3.2. LncPLAAT3 Promotes Porcine Primary Preadipocyte Differentiation

To assess the role of lnc*PLAAT3*-AS in porcine preadipocyte differentiation, we transfected porcine primary preadipocytes with one of the three constructs described in Section 3.1 and then induced lipogenic differentiation for 8 days. As shown in Figure 1E, oil red O staining demonstrated that inhibition of lnc*PLAAT3* expression significantly reduced lipid droplet formation compared with the negative control group, while overexpression of lnc*PLAAT3*-AS had the opposite effect. To confirm these results, we examined the expression levels of *CEBPα*, *PPARγ*, and *FABP4*, which are markers of adipocyte differentiation [17]. We found that transfection with the lnc*PLAAT3*-AS overexpression vector significantly enhanced the expression of all three of these genes, while transfection with the siRNA inhibited their expression (Figure 1F). These suggest that lnc*PLAAT3*-AS promotes adipocyte differentiation. In addition, we examined the expression of genes related to fatty acid synthesis (*ACS* and *ACADL*) [18] and fatty acid oxidation (*DGAT* and *FAS*). As expected, *ACS* and *ACADL* expression were significantly increased and *DGAT* and *FAS* expression were significantly decreased in the lnc*PLAAT3* overexpression group compared to the control group, while the opposite effect was observed in the siRNA-treated group (Figure 1F). These experimental results were further validated at the protein level within the samples (Figure 1G). Based on the above results, it is reasonable to conclude that lnc*PLAAT3*-AS promotes preadipocyte differentiation.

### 3.3. Differences in Gene Expression in Adipocytes after lncPLAAT3 Overexpression or Knockdown

To characterize the regulatory role of lnc*PLAAT3*-AS in primary preadipocytes, we performed RNA-seq analysis of cells in which lnc*PLAAT3*-AS was overexpressed or knocked down, as well as the negative control, with each sample yielding an average of 6.72 G of data (Appendix A). The correlation coefficient and principal component analyses showed a clear separation among the three groups (Figure 2A,B). Compared with the negative control group, there were 824 differentially expressed genes (DEGs) in the lnc*PLAAT3*-AS overexpression group and 2219 DEGs in the lnc*PLAAT3*-AS siRNA group (Figure 2C,D, Appendix A). The potential functions and signaling pathways of all DEGs were determined by GO and KEGG enrichment analyses. Upregulated genes in the lnc*PLAAT3*-AS overexpression group compared with the control group (such as *ADORA1*, *CCR5*, and *CD36*) were associated with an inflammatory response, cellular response to lipid, and regulation of lipid storage; downregulated genes in the lnc*PLAAT3*-AS overexpression group compared with the control group (such as *CSF2, TFPI*, and *NOS3*) were associated with a cellular response to lipopolysaccharide, lipid homeostasis, and positive regulation of lipid localization. Upregulated genes in the lnc*PLAAT3*-AS siRNA group compared with the control group (such as *ABL2, ADCY1,* and *ARG1*) were associated with a response to lipopolysaccharide and Semaphorin-plexin signaling, while downregulated genes in the lnc*PLAAT3*-AS siRNA group compared with the control group (such as *ACTC1, ACTN2,* and *ADGRB1*) were associated with cellular component morphogenesis (Figure 2E–H and Appendix A). Interestingly, 20 genes closely related to fat metabolism, such as *CD36*, *CD68*, *CCR5*, and *TREM2*, exhibited the same expression patterns as *PLAAT3,* indicating that *PLAAT3* plays an important role in adipogenesis (Figure 2I,J) [19,20,21,22]. For example, the scavenger receptor *CD36* participates in the high-affinity tissue uptake of long-chain fatty acids (FAs) and contributes to lipid accumulation and metabolic dysfunction in excess [18].

### 3.4. LncPLAAT3-AS Targets miR-503-5p during Adipocyte Differentiation

Numerous studies have shown that lncRNAs can regulate various cellular responses by sponging on miRNAs [3,4,6,23]. Therefore, we used a lncTar online prediction software (http://www.cuilab.cn/lnctar (accessed on 23 November 2022)) and a dual-luciferase reporter system to screen for candidate miRNA binding partners and identified miR-503-5p as a potential lnc*PLAAT3*-AS target gene (Figure 3A). To determine whether this was an authentic interaction, we assayed the chemiluminescence values of PK15 cells cotransfected with the wild-type and mutant lnc*PLAAT3*-AS vectors described above and an miR-503-5p mimic or an NC mimic, using an enzyme marker. As expected, the relative fluorescence values of the cells cotransfected with the miR-503-5p mimic and the lnc*PLAAT3*-AS-WT vector were significantly lower than those of the other three groups (Figure 3B,C). These results validated the target gene prediction for lnc*PLAAT3*-AS. Moreover, we verified the interaction between miR-503-5p and *PLAAT3* using a dual-luciferase reporter system and found that miR-503-5p targets *PLAAT3* (Figure 3D). Next, we asked whether miR-503-5p is involved in porcine preadipocyte proliferation and differentiation. Then, we performed the mock transfection of porcine preadipocytes or transfected them with miR-503-5p and mock NC, inhibitor, or inhibitor NC (Figure 3E) to examine whether miR-503-5p affects preadipocyte proliferation. Transfection with the miR-503-5p mimics significantly inhibited preadipocyte proliferation, while the inhibitor-treated group had the opposite response (Figure 3F). CCK-8 and EdU assays confirmed this observation (Appendix A), as did fluorescence quantification of the expression of marker genes for proliferation and cell cycle progression (Appendix A). We also analyzed the effect of transfection of porcine preadipocytes with an miR-503-5p mimic, an inhibitor, and separate control reagents on preadipocyte differentiation. Surprisingly, miR-503-5p overexpression significantly inhibited the ability of preadipocytes to secrete lipid droplets, while cells transfected with the inhibitor had the opposite response (Appendix A). Analysis of the expression of marker genes for lipogenic differentiation and fatty acid metabolism confirmed this result (Appendix A). Taken together, these findings suggest that, in addition to enhancing the stability of *PLAAT3* by forming an RNA–RNA dimer [11], lnc*PLAAT3*-AS also helps to regulate adipogenesis and related molecules through miR-503-5p.

### 3.5. Differences in lncPLAAT3-AS and PLAAT3 Proliferation among Different Pig Breeds

To explore the role of lnc*PLAAT3*-AS and *PLAAT3* in fat deposition among different pig breeds, we downloaded the RNA-seq data for seven Chinese pig breeds, namely Chenghua, Neijiang, Qingyu, Yanan, Wujin, Yacha, and Tibetan, as well as one introduced Western breed, Yorkshire, with three biological replicates per breed. We analyzed *PLAAT3* and lnc*PLAAT3*-AS expression among the different breeds and found that both showed slight upregulation, though statistically insignificant, in six of the seven Chinese pig breeds (except for Wujin) compared with the Western breed, Yorkshire. This suggests that both *PLAAT3* and lnc*PLAAT3*-AS play an essential role in fat deposition in pigs and are potentially involved in the distinct adiposity phenotype between Chinese (relatively obese) and Western (relatively lean) breeds (Figure 4A,B).

### 3.6. SNPs of lncPLAAT3-AS and PLAAT3

Single nucleotide polymorphic SNPs may disrupt the structural features of many non-coding RNAs, interfere with their molecular function, and produce phenotypic effects [24]. We therefore searched for and identified a variety of SNP sites within lnc*PLAAT3*-AS (Appendix A). Although these SNPs do not occur in the region of lnc*PLAAT3*-AS that binds to miR-503-5p, they may change the secondary structure and/or expression of lnc*PLAAT3*-AS, which should be explored in future studies.

## 4. Discussion

In this study, we found that lnc*PLAAT3*-AS promotes adipocyte lipogenic differentiation and mediates the miR-503-5p/*PLAAT3* interaction in adipocytes. The knockdown of lnc*PLAAT3*-AS in adipocytes decreases cell viability and inhibits adipocyte proliferation and lipogenic differentiation, whereas the knockdown of miR-503-5p had the opposite effect [25,26]. Our results suggest that lnc*PLAAT3*-AS and miR-503-5p could serve as targets for manipulating adipocyte lipogenic differentiation.

Based on previous studies of lnc*PLAAT3*-AS, we investigated the role of lnc*PLAAT3*-AS in regulating porcine primary preadipocyte development by overexpressing or disrupting this lncRNA in porcine primary preadipocytes and analyzing the effects with RNA-seq technology. In addition to enhancing *PLAAT3* stability by forming RNA–RNA dimers, it appears that lnc*PLAAT3*-AS functions within a second regulatory pathway to regulate adipogenesis and related molecules. One of the most well-accepted models for lncRNAs function is their role as ceRNAs which act as sponges to absorb free miRNAs via sequence complementarity, thereby inhibiting target miRNA function [27]. Li et al. reported that *UBE2CP3* promotes gastric cancer development mainly through the miR-138-5p/*ITGA2* axis. In addition, their study showed that *UBE2CP3*/*IGFBP7* can form an RNA duplex that interacts directly with the ILF3 protein. ILF3-mediated RNA–RNA interactions between *IGFBP7* mRNA and *UBE2CP3* in turn play an important role in protecting the stability of *UBE2CP3* mRNA [23]. We found that lnc*PLAAT3*-AS significantly promotes preadipocyte proliferation and differentiation by targeting miR-503-5p. Therefore, our findings suggest that lnc*PLAAT3*-AS plays an essential role in lipid deposition in pigs.

Adipocyte differentiation is essential for lipid deposition in mammals [28]. It has been shown that the normal expression of the *PLAAT3* gene promotes lipid deposition in mice. However, the rate of lipolysis was significantly higher in *PLAAT3*-deficient mice due to the significantly lower expression of adipose prostaglandin E2 bound to the Gai-coupled receptor EP3, increased cyclic AMP expression, and significantly lower adipose tissue mass [10,11]. In our study, we performed RNA-seq on porcine preadipocytes, followed by both GO and KEGG enrichment analysis, and found that lnc*PLAAT3*-AS is closely related to cellular metabolism, as is its host gene, *PLAAT3*. Further analysis showed that the expression changes of many critical genes involved in fat metabolism mirrored those of *PLAAT3*. The cluster of differentiation 36 (*CD36*) is the upstream gene of *PLAAT3*, whose main function is to release arachidonic acid (AA) from the cell membrane via cytoplasmic phospholipase A(2)α (cPLA(2)α), which is one of the products of *PLAAT3*. In addition to this, *CD36* contributes to the production of pro-inflammatory eicosanoids. Compared to control cells, CHO cells normally expressing human *CD36* released significantly more AA and prostaglandin E(2) (PGE(2)) in response to thapsigargin-induced ER stress [29]. The results in this study are worthy of further testing in a more significant number of samples.

British Yorkshire and local Chinese pigs differ significantly in meat production traits and have complementary characteristics. Local Chinese pigs have tender, flavorful, and juicy meat but grow slowly and produce little lean meat [30] Although Yorkshire pigs grow faster and produce more lean meat, the quality of the meat is not as high as that of local Chinese breeds. The differences in *PLAAT3* and lnc*PLAAT3*-AS expression among the eight pig breeds are also consistent with the abovementioned differences in phenotypic traits, demonstrating that *PLAAT3* and lnc*PLAAT3*-AS play an important role in pig fat deposition [15,16].

Increasingly, high-throughput sequencing has been used to identify non-coding RNAs involved in adipogenesis [31,32]. Non-coding RNAs, which were once considered nonsense transcripts, play a rich variety of essential roles in various biological processes. Therefore, it is crucial to continue to explore the diversity of non-coding RNAs and validate their functions to advance our understanding of epigenetic regulation and gain a deeper appreciation of the range of genetic information carried by these molecules [33].

## 5. Conclusions

In summary, we report the role of the *PLAAT3*–lnc*PLAAT3*-AS–miR-503-5p regulatory axis in preadipocyte development. lnc*PLAAT3*-AS promotes the expression of the host gene *PLAAT3* by adsorbing miR-503-5p, thereby promoting the lipogenic differentiation of porcine primary preadipocytes. Furthermore, we identified related genes that may be involved in adipogenesis by performing RNA-seq on preadipocytes in which lnc*PLAAT3*-AS was overexpressed or inhibited and explored the differences in *PLAAT3* and lnc*PLAAT3*-AS expression among different pig breeds. Our study provides a theoretical basis for other researchers to explore the molecular mechanisms of non-coding RNA-mediated regulation of lipid deposition.

## Figures and Tables

**Figure 1 genes-14-00161-f001:**
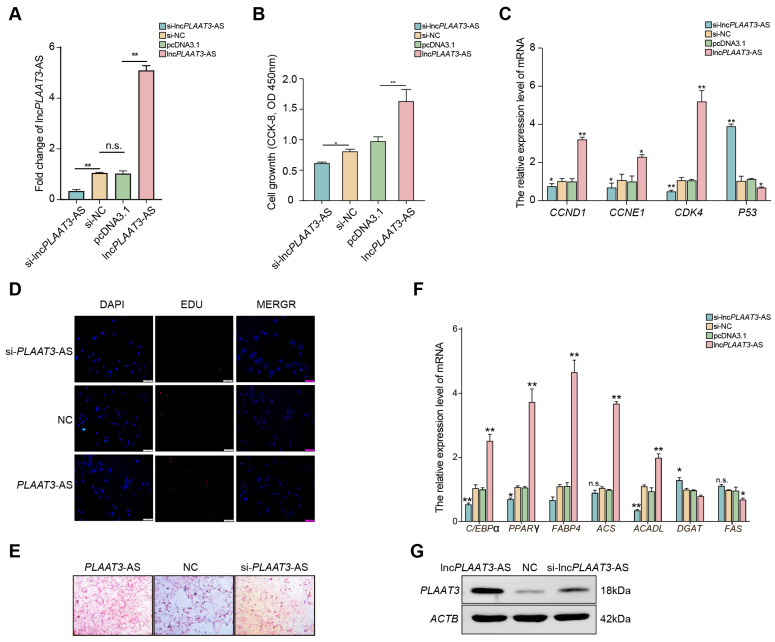
Lnc*PLAAT3*-AS promotes preadipocyte proliferation and differentiation. Primary adipocytes were transfected with a lnc*PLAAT3*-AS overexpression plasmid, an siRNA against lnc*PLAAT3*-AS (si-lnc*PLAAT3*-AS), or a negative control (si-NC) construct; (**A**) transfection efficiency was measured by qRT-PCR; (**B**) cell proliferation was evaluated by CCK-8 assay; (**C**) expression of genes associated with cell proliferation was measured by qRT-PCR; (**D**) representative images of an EdU assay of PK15 cells transfected with lnc*PLAAT3*-AS mimics or lnc*PLAAT3*-AS inhibitor; (**E**) oil red O staining; (**F**) expression of marker genes related to adipogenesis, fatty acid oxidation, and fatty acid transportation synthesis (scale bars 100 μm); (**G**) *PLAAT3* expression after transfection with each of the three constructs described above. All results are presented as means ± SEM; n = 3; * *p* < 0.05; ** *p* < 0.01; n.s., no significant difference.

**Figure 2 genes-14-00161-f002:**
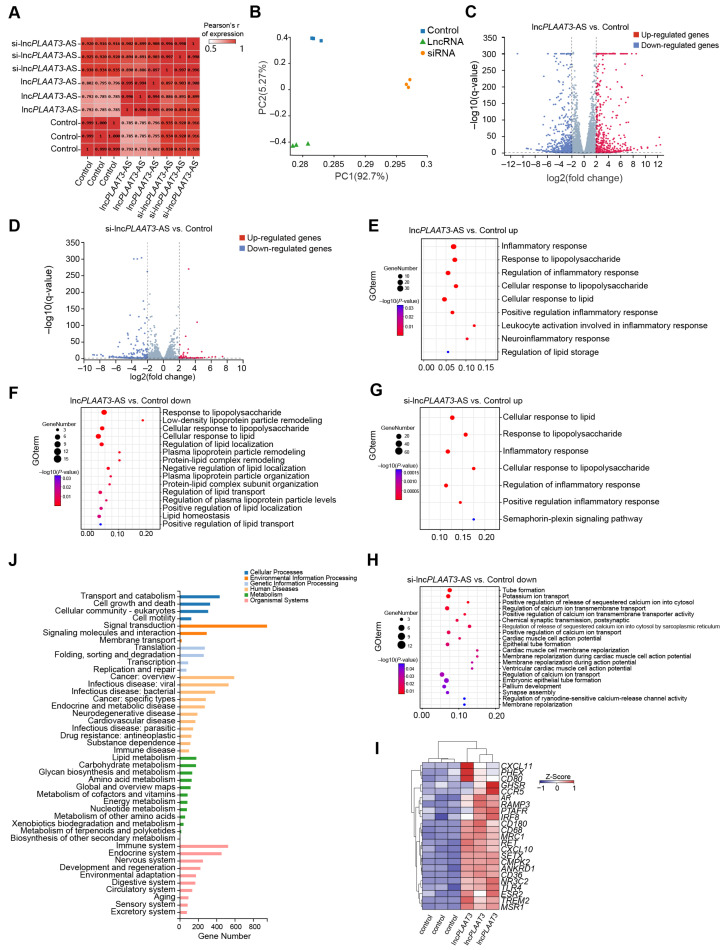
(**A**) Sample correlation heatmap; (**B**) principal component analysis; (**C**,**D**) volcano plot of differentially expressed genes. Positive values indicate upregulation, while negative values indicate downregulation. The x-axis represents the log2 scale of fold change. A significant difference in expression is indicated by the y-axis, which is the −log10 scale of the adjusted *p* values. The red dots in the figure indicate genes with significantly upregulated expression (at least two-fold change), blue dots indicate genes with significantly downregulated expression (at least two-fold change), and gray dots indicate genes with no significant difference in expression level. (**E**–**H**) The top 20 GO pathways for the differentially expressed genes according to the following comparisons: upregulated in lnc*PLAAT3*-AS vs. control, downregulated in lnc*PLAAT3*-AS vs. control, upregulated in si-lnc*PLAAT3*-AS vs. control, downregulated in si-lnc*PLAAT3*-AS vs. control; (**I**) differentially expressed gene pathway enrichment analysis; (**J**) All enriched GO and KEGG pathways.

**Figure 3 genes-14-00161-f003:**
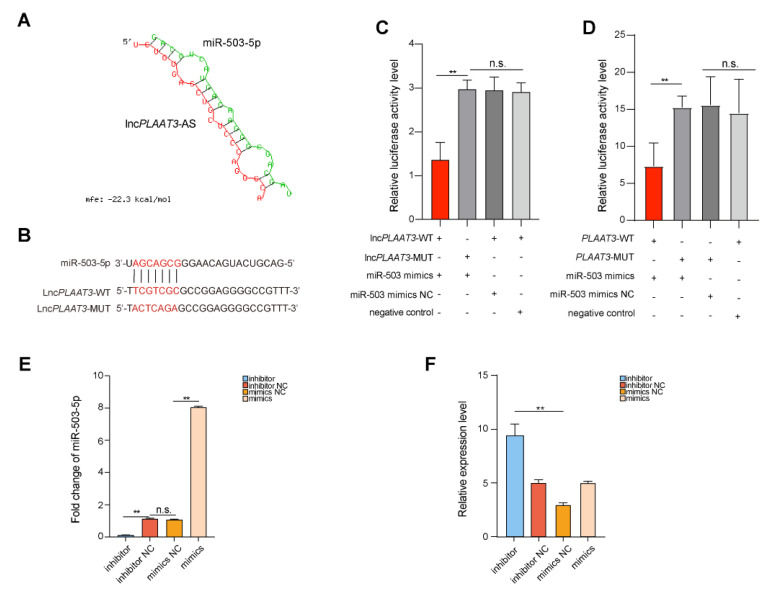
(**A**) Predicted miR-503-5p binding site in lnc*PLAAT3*-AS as determined by RNAhybrid; (**B**) binding capacity of lnc*PLAAT3*-AS and miR-503-5p as evaluated by RNAhybrid (Red sequences are miRNA seed region binding sequences); (**C**) luciferase assay revealed miR-503-5p-mediated inhibition of lnc*PLAAT3*-AS activity, miR-503-5p mimics NC is miR-503-5p mimics negative control (“+” indicates transfected, “−” indicates not transfected); (**D**) luciferase assay revealed miR-503-5p–mediated inhibition of *PLAAT3* activity; (**E**) Transfection efficiency as measured by qRT-PCR, miR-503-5p inhibitor NC is miR-503-5p inhibitor negative control; (**F**) qRT-PCR analysis of *PLAAT3* expression in transfected cells. All results are presented as means ± SEM; n = 3; ** *p* < 0.01; n.s., no significant difference.

**Figure 4 genes-14-00161-f004:**
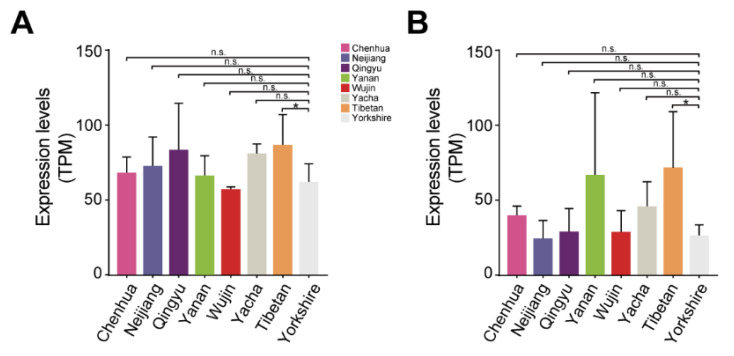
(**A**) Bar graph showing *PLAAT3* expression in eight pig breeds. (**B**) Bar graph showing lnc*PLAAT3*-AS expression in eight pig breeds. All results are presented as means ± SEM; n = 3; * *p* < 0.05; n.s., no significant difference.

**Table 1 genes-14-00161-t001:** Primer information for quantitative real-time PCR (qRT-PCR).

Gene	Primer Sequence (5′ to 3′)	Product Size (bp)
*PLA2G16*	F: ATATGTGGTCCACCTGGCTCCCC	395
	R: ATTGCTTTTGCCGCTTGTTTCTG	
*PLA2G16-AS*	F: GGACTCTGCGGCCATTTAAC	213
	R: GCTTTGGGACAATGAGTCGC	
*GAPDH*	F: CCCCTTCATTGACCTCCACT	192
	R: CCATTTGATGTTGGCGGGAT	
*CCND1*	F: GCATGTTCGTGGCCTCTAAG	228
	R: CGTGTTTGCGGATGATCTGT	
*CDK4*	F: TCAGCACAGTTCGTGAGGTG	77
	R: GTCCATCAGCCGGACAACAT	
*P53*	F: GGGACGGAACAGCTTTGA	161
	R: TTTGCACTGGCGAGGAG	
*PPARγ*	F: CTCCAAGAATACCAAAGTGCGA	150
	R: GCCTGATGCTTTATCCCCACA	
*C/EBPα*	F: CAAGAACAGCAACGAGTACCG	124
	R: GTCACTGGTCAACTCCAGCAC	
*FABP4*	F: GAAGTGGGAGTGGGCTTT	190
	R: TTATGGTGCTCTTGACTTTCCT	
*ACS*	F: GCAGGCAGGCTCAGTTT	129
	R: CTCTGTTCAGGGGAGGGT	
*ACADL*	F: TGTCTCCAGCTGCATGAAACGA	107
	R: AGCTGCACACAGTCATAAGCCA	
*DGAT*	F: CCTACCGCGATCTCTACTACTT	126
	R: GGGTGAAGAACAGCATCTCAA	
*FAS*	F: CCAACCAGCAACACCAA	100
	R: CAGGTACGGGAATGAGGA	
ssc-miR-503	UAGCAGCGGGAACAGUACUGCAG	
U6	F: CGCTTCGGCAGCACATATAC	87
	R: TTCACGAATTTGCGTGTCAT	

## Data Availability

The RNA-seq data is available in the NCBI Gene Expression Omnibus (GEO; https://www.ncbi.nlm.nih.gov/geo/, accessed on 26 September 2022) under series numbers GSE213924.

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
