# Peer review of "LncPLAAT3-AS Regulates PLAAT3-Mediated Adipocyte Differentiation and Lipogenesis in Pigs through miR-503-5p"

_genes, 2023, doi:10.3390/genes14010161_

Round 1
Reviewer 1 Report
The paper:"LncPLAAT3-AS regulates PLAAT3-mediated adipocyte differentiation and lipogenesis in pigs through miR-503-5p" describes an experiment in which the authors try to prove the role of LNC PLAAT3-AS in the differentiation of adipose tissue in pigs by overexpressing or knocking down lncPLAAT3-AS in porcine primary preadipocytes and by reanalyzing gene expression data from lean and fatty pig breeds. The experimental approach is correct but the methods are poorly described. Moreover the manuscript is inaccurate (lack of supplementary figures, some strange, inserted sentences (line 73). More specific comments:
line 15: I am not sure if "lipogenic differentation" is correct
line 36 NqO - change to italics
line 37 CS -induced - please explain
line 73 a device ? did you mean "centrifuged"?
line 118oligo dt and random primers were used at the same time?
lines139-145 - please provide some references for DNA nano balls technology and describe how cDNA with adapters were obtained.
line 193 n=3 is it a biological replicate or technical - if biological please explain at wich stage of the experiment?
line 296 how many individuals per breed were analyzed? please provide the accession numbers and the database for downloadede RNA-seq data
line 300 - fig 4 caption - pig species? please correct
line 318 GC?
line 350 it should be mentioned in the results section
supplementary figure: Please explain why the mapping rate was so low - 44%
supplementary table please provide p-values
I could not find more supplementary materials
Reviewer 2 Report
Pigs are model animals that are frequently used in medical research. The topic is relevant.
The article may be accepted after making minor corrections.
1. In the Abstract section mention clearly the main aim of the work.
2. L 69: “Fat tissue from the backs of the female Rongchang piglets were collected to obtain preadipocytes” ; were collected under anaesthesia?, what was the sample size?
3. “fat tissue was quickly minced” what was the sample size?
4. L 73: “The digested tissue solution was then (a device that spins something at high speed) at 1000 rpm for 8 minutes”. correct “…tissue solution was then centrifuged…”
5. L 295: „We analyzed PLAAT3 … expression among the different breeds and found that both were upregulated in the seven Chinese pig breeds compared with the Western breed Yorkshire,…” compared with the Wujin breed ?; the Figure 4 does not confirm this.
6. The Results section contains the results of a study and discussion. I propose to combine the results with the discussion.
7. Section of References : Make corrections in accordance with the guidelines for authors, e.g. when the middle name is written as an initial, it should be capitalized, some references are missing journal name, provide abbreviations of the titles of the journals.
Round 2
Reviewer 1 Report
I believe that nanoballs technology still needs further explanation ,and three technical replicates it is not enough to to draw a strong conclusions
